# Activation of the NRF2 antioxidant program generates an imbalance in central carbon metabolism in cancer

Volkan I Sayin[1†], Sarah E LeBoeuf[1†], Simranjit X Singh[1], Shawn M Davidson[2,3], Douglas Biancur[1], Betul S Guzelhan[1], Samantha W Alvarez[1], Warren L Wu[1], Triantafyllia R Karakousi[1], Anastasia Maria Zavitsanou[1], Julian Ubriaco[1], Alexander Muir[2], Dimitris Karagiannis[1], Patrick J Morris[4,5], Craig J Thomas[4,5], Richard Possemato[1], Matthew G Vander Heiden[2,3], Thales Papagiannakopoulos[1*]

[1]Department of Pathology, New York University School of Medicine, New York, United States; [2]Koch Institute for Integrative Cancer Research, Massachusetts Institute of Technology, Cambridge, United States; [3]Department of Biology, Massachusetts Institute of Technology, Cambridge, United States; [4]NIH Chemical Genomics Center, National Center for Advancing Translational Sciences, Bethesda, United States; [5]Division of Preclinical Innovation, National Center for Advancing Translational Sciences, Bethesda, United States

**\*For correspondence:**
papagt01@nyumc.org

[†]These authors contributed equally to this work

**Abstract** During tumorigenesis, the high metabolic demand of cancer cells results in increased production of reactive oxygen species. To maintain oxidative homeostasis, tumor cells increase their antioxidant production through hyperactivation of the NRF2 pathway, which promotes tumor cell growth. Despite the extensive characterization of NRF2-driven metabolic rewiring, little is known about the metabolic liabilities generated by this reprogramming. Here, we show that activation of NRF2, in either mouse or human cancer cells, leads to increased dependency on exogenous glutamine through increased consumption of glutamate for glutathione synthesis and glutamate secretion by $x_c^-$ antiporter system. Together, this limits glutamate availability for the tricarboxylic acid cycle and other biosynthetic reactions creating a metabolic bottleneck. Cancers with genetic or pharmacological activation of the NRF2 antioxidant pathway have a metabolic imbalance between supporting increased antioxidant capacity over central carbon metabolism, which can be therapeutically exploited.
DOI: https://doi.org/10.7554/eLife.28083.001

## Introduction

Cancer cells rewire their metabolism to sustain uncontrolled cellular proliferation (*Hanahan and Weinberg, 2011*; *Vander Heiden and DeBerardinis, 2017*). Alterations to cellular metabolism are required to meet the increased energetic and biosynthetic demands of cancer cells (*DeBerardinis and Chandel, 2016*; *Vander Heiden and DeBerardinis, 2017*). Warburg first observed that cancer cells have an increased demand for glucose and increased glycolysis compared to non-cancer cells (*Warburg et al., 1927*). It is now increasingly clear that during transformation and tumorigenesis cancer cells require additional metabolic alterations, which result in dependencies on exogenous metabolites such as the non-essential amino acid glutamine. Although cells can readily synthesize glutamine, several studies have demonstrated that some cancer cells rely on exogenous glutamine and require supplementation for survival (*Lacey and Wilmore, 1990*; *DeBerardinis et al., 2007*; *DeBerardinis and Cheng, 2010*; *Wise and Thompson, 2010*; *Altman et al., 2016*; *Vander Heiden and DeBerardinis, 2017*). Glutamine is the most abundant

amino acid found in blood (*Mayers and Vander Heiden, 2015*) and can be used for a variety of purposes including energy production and generation of biosynthetic intermediates by fueling the TCA cycle (*DeBerardinis et al., 2007*; *Wise and Thompson, 2010*; *Lane and Fan, 2015*; *Hosios et al., 2016*). Glutaminase catalyzes the conversion of glutamine to glutamate and is the rate-limiting enzyme for glutaminolysis (*DeBerardinis et al., 2007*; *Deberardinis et al., 2008*; *DeBerardinis and Cheng, 2010*; *Altman et al., 2016*; *Davidson et al., 2016*). Glutamine-derived glutamate is also critical for the generation of the most abundant cellular metabolite, the antioxidant glutathione (GSH), making glutamine metabolism important for maintaining redox homeostasis (*Welbourne, 1979*; *Son et al., 2013*; *Xiang et al., 2013*). Numerous oncogenes drive increased glutamine metabolism, hence targeting glutaminolysis in particular genetic subtypes of cancer has emerged as an attractive therapeutic option (*Yuneva et al., 2007*; *Gao et al., 2009*; *Hu et al., 2010*; *Gaglio et al., 2011*; *Wise et al., 2011*; *Yuneva et al., 2012*; *Son et al., 2013*; *Qie et al., 2014*). Despite extensive studies on the role of glutamine in KRAS-driven tumors (*Cox et al., 2014*), inhibiting this pathway has led to limited responses in both pre-clinical models (*Davidson et al., 2016*; *Biancur et al., 2017*) and ongoing clinical trials; highlighting the importance of identifying genetic subtypes of cancer that may have a greater dependency on glutamine.

During tumor initiation and progression, increased metabolic output as well as changing environmental conditions result in high oxidative stress in the form of reactive oxygen species (ROS) (*Cairns et al., 2011*). The role of ROS in the development and progression of cancer has been the subject of considerable study and debate (*Chandel and Tuveson, 2014*; *Chio and Tuveson, 2017*). Emerging evidence supports the idea that cancers increase antioxidant capacity as a stress response mechanism, suggesting that high ROS levels may constitute a barrier to tumor progression (*Sayin et al., 2014*; *Le Gal et al., 2015*; *Piskounova et al., 2015*). To maintain oxidative homeostasis, cancer cells increase their antioxidant capacity (*Trachootham et al., 2009*). Under normal physiologic conditions, the KEAP1/NRF2 signaling axis is responsible for coordinating the cell's antioxidant defenses in response to ROS. KEAP1 binds to NRF2 and enables the CUL3-mediated ubiquitination and proteasomal degradation of NRF2 (*Kobayashi et al., 2004*). When ROS increase, KEAP1 oxidation leads to a conformational change enabling the dissociation and stabilization of NRF2, which translocates to the nucleus to induce the transcription of a battery of antioxidant genes (*Itoh et al., 1997*; *Itoh et al., 1999*), including enzymes regulating the synthesis of the major cellular antioxidant GSH, and enzymes involved in the reduction of oxidized GSH and thioredoxin (TXN)(*Gorrini et al., 2013*). The activation of the antioxidant response requires the flux of glucose and glutamine-derived carbons and cofactors (e.g. NAD/NADH, NADP/NADPH) to be diverted to antioxidant production (*Mitsuishi et al., 2012*; *Singh et al., 2013*). Increased flux of nutrients to antioxidant production may limit nutrient availability for other biosynthetic pathways.

However, in the context of chronic ROS exposure, post-transcriptional or post-translational signaling can stabilize NRF2 (*Goldstein et al., 2016*), or epigenetic (*Muscarella et al., 2011*; *Hanada et al., 2012*; *Fabrizio et al., 2017*) and genetic events can lead to NRF2 stabilization (*Shibata et al., 2008a*; *Kim et al., 2010*; *Konstantinopoulos et al., 2011*; *Jaramillo and Zhang, 2013*; *Sato et al., 2013*), resulting in the chronic activation of NRF2 and its downstream antioxidant program (*Nioi and Nguyen, 2007*; *Shibata et al., 2008a*; *Shibata et al., 2008b*). Interestingly, *KEAP1* is the third most frequently mutated gene in lung adenocarcinoma (LUAD) and often co-occurs with oncogenic mutations in *KRAS* (*Cancer Genome Atlas Research Network, 2014*). Activation of the NRF2-driven antioxidant axis may create a unique set of metabolic requirements necessary to sustain this increased antioxidant capacity (*Mitsuishi et al., 2012*; *DeNicola et al., 2015*; *Koppula et al., 2017*), creating potential for novel therapeutic vulnerabilities in aggressive lung cancers.

Here we demonstrate that *KEAP1* loss of function (LOF) mutations drive increased dependency on glutamine in both mouse and human KRAS-driven LUAD cell lines. We show that *KEAP1* mutant cells have decreased intracellular glutamate pools through increased glutamate consumption for GSH synthesis and by exporting glutamate through the antiporter xCT in exchange for cystine. The low intracellular pools of glutamate lead to increased sensitivity to glutamine deprivation and glutaminase inhibition in an xCT dependent fashion. Using a small molecule activator of NRF2, we provide evidence that acute NRF2 activation is sufficient to rewire cellular metabolism, similar to that of *KEAP1* mutant cells, and leads to glutamine dependency due to a basal deficiency in anaplerosis. Finally, we show that this is a phenomenon that occurs across multiple types of cancers with *KEAP1*

mutations, and demonstrate the importance of sub-stratifying patients based on genotype to maximize therapeutic efficacy of glutaminase inhibition in clinical trials and pre-clinical models where responses have been previously limited (*Davidson et al., 2016*; *Biancur et al., 2017*).

## Results

### KEAP1 mutations cause increased dependency on exogenous glutamine

To study the role of *Keap1* mutations in rewiring lung cancer metabolism, we generated isogenic Kras-driven, *Trp53* null (Kras$^{G12D/+}$; p53$^{-/-}$; hereafter KP) cells with wild-type (KP) or LOF mutations in *Keap1* (KPK) using CRISPR/Cas9-editing. We observed that KPK cells had increased production of GSH compared to KP cells (*Figure 1A*), as a result of increased levels of glutamate-cysteine ligase catalytic subunit, *Gclc*, which along with glutamate-cysteine ligase modifier subunit, *Gclm* synthesize GSH (*Figure 1—figure supplement 1A and B*). As expected, elevated levels of GSH in KPK cells

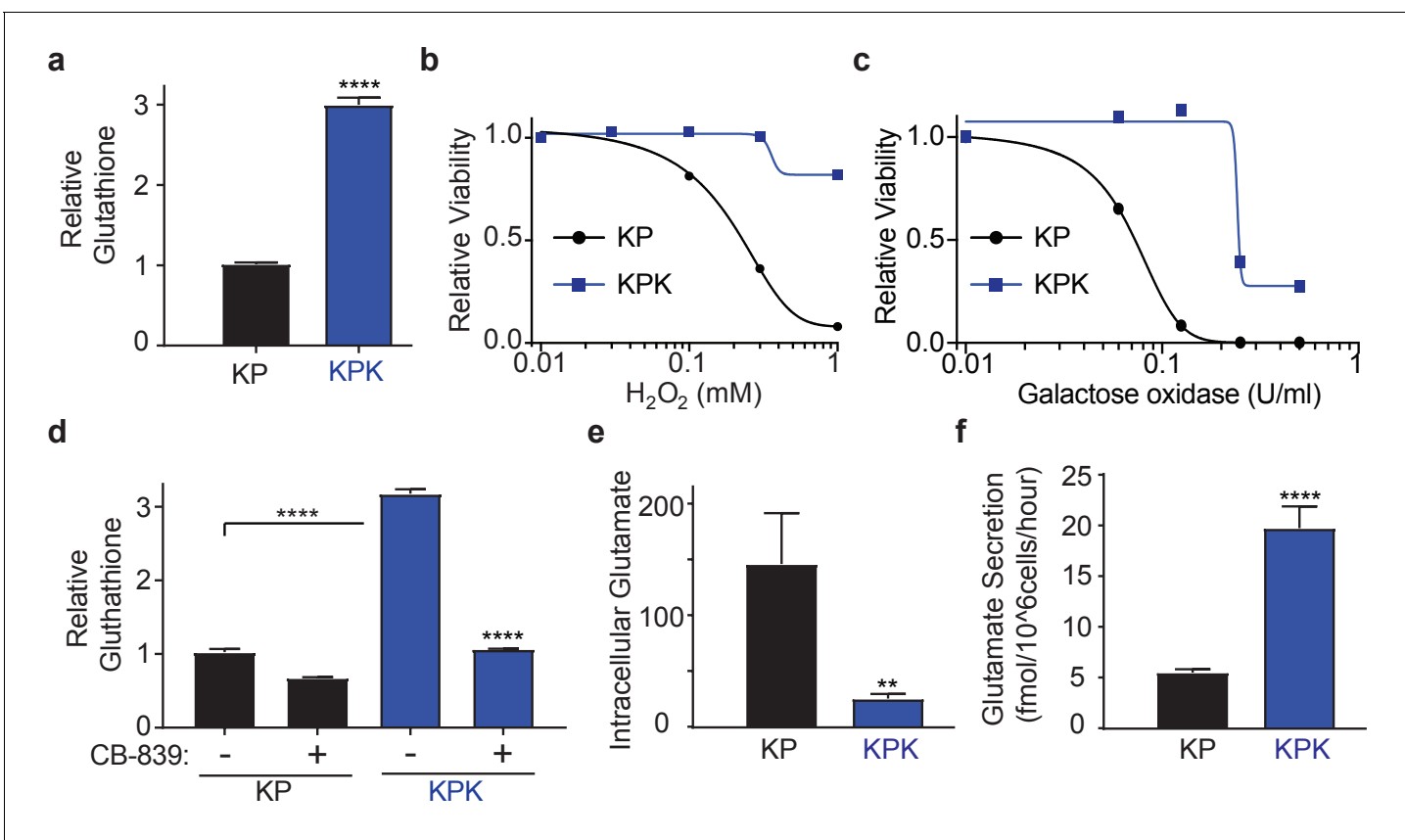

**Figure 1.** *Keap1* mutations cause increased dependency on exogenous glutamine. (a) Measurement of whole cell glutathione levels in wild type (KP) and *Keap1* mutant (KPK) in isogenic clones derived from mouse lung tumors (*n* = 3, triplicate wells). (b) Relative viability assayed with cell-titer glo (relative luminescent units) in KP and KPK cells in RPMI after treatment with H$_2$O$_2$ for 12 hr. All data points are relative to vehicle treated controls (*n* = 8/data point from replicate wells) (c) Relative viability assayed with cell-titer glo (relative luminescent units) in KP and KPK cells in RPMI after treatment with galactose oxidase for 72 hr. All data points are relative to vehicle treated controls (*n* = 4/data point from replicate wells). d) Measurement of whole cell glutathione levels in KP and KPK cells after 24 hr of CB-839 treatment where indicated (*n* = 3, triplicate wells). (e) Intracellular glutamate levels per cell (*n* = 6, two independent experiments with triplicate wells). (f) Glutamate secretion in KP and KPK cells (*n* = 6, two independent experiments with triplicate wells). All error bars depict s.e.m. **p<0.01, ****p<0.0001.

DOI: https://doi.org/10.7554/eLife.28083.002

The following figure supplement is available for figure 1:

**Figure supplement 1.** *Keap1* mutation increases cellular antioxidant capacity and sensitizes KPK cells to glutamine deprivation or glutaminase inhibition.

DOI: https://doi.org/10.7554/eLife.28083.003

corresponded with decreased levels of cellular ROS (*Figure 1—figure supplement 1C*). Both KP and KPK cells had similar growth rates under basal conditions (*Figure 1—figure supplement 1D*), however when challenged with oxidative stress, KPK cells were more resistant compared to KP cells (*Figure 1B and C*, *Figure 1—figure supplement 1E*).

GSH is a tri-peptide composed of glutamate, cysteine, and glycine. Therefore, we conjectured that high levels of GSH in KPK cells might result in increased demand for glutamate and consequently KPK cells would be more dependent on glutamine-derived glutamate compared to KP cells. Indeed, decreasing glutamine concentrations in the media substantially decreased proliferation of KPK cells with minimal effects on KP cells (*Figure 1—figure supplement 1F*). Glutaminase catalyzes the conversion of glutamine to glutamate and is the rate-limiting enzyme for glutaminolysis (*DeBerardinis et al., 2007*; *DeBerardinis and Cheng, 2010*; *Altman et al., 2016*; *Davidson et al., 2016*). Here we use a small molecule inhibitor of glutaminase, CB-839, which is currently in clinical phase I trials for multiple cancer types, including LUAD. CB-839 blocks glutamine catabolism and decreases the availability of glutamate in the cell for GSH antioxidant production and the generation of α-ketoglutarate (αKG) as a carbon source for the TCA cycle and other biosynthetic intermediates (*Altman et al., 2016*). Consistent with glutamine deprivation, KPK cells had increased sensitivity to CB-839, (*Figure 1—figure supplement 1G*). In line with CB-839 decreasing glutamate availability, cellular glutathione levels were reduced after CB-839 treatment (*Figure 1D*). Increased sensitivity to both glutamine deprivation as well as glutaminase inhibition suggests that *Keap1* mutant lung cancers depend on glutamine-derived glutamate to support GSH synthesis.

## xCT/Slc7a11-dependent glutamate secretion in *Keap1* mutant cells causes glutamine dependency

To elucidate the mechanism of glutamine dependency in *Keap1* mutant cells resulting from increased cellular glutamate demand, we measured both intra- and extra-cellular glutamate levels. Surprisingly, KPK cells had lower intracellular but higher extracellular levels of glutamate when compared to KP cells (*Figure 1E and F*). Glutamate is required for the synthesis of GSH, but is also necessary for the import of cystine via the $x_c^-$ antiporter system (xCT), which exchanges glutamate for cystine across the plasma membrane in order to support intracellular cysteine pools for antioxidant production (*Figure 2A*) (*Lewerenz et al., 2013*). Previously, expression of xCT has been linked with glutamine sensitivity and to antagonize glutamine anaplerosis (*Timmerman et al., 2013*; *Shin et al., 2017*). xCT is a heterodimer of Slc7a11, a *bona-fide* Nrf2 target gene, and Slc3a2 (*Lewerenz et al., 2013*). Consistent with prior findings (*Ishii et al., 1987*; *Lewerenz et al., 2013*; *Habib et al., 2015*), gene expression analysis revealed dramatically higher levels of Slc7a11 in KPK cells compared to KP controls (*Figure 2B*). We hypothesized that the Nrf2-driven increase in expression of *Slc7a11* and *Gclc*, and higher levels of GSH in KPK cells may result in export of glutamate in exchange for import of cystine (*Figure 2A*). To determine if xCT/Slc7a11 is responsible for the KPK cell dependency on exogenous glutamine, we treated cells with a small molecule inhibitor of xCT, Erastin (*Dixon et al., 2014*). Indeed, Erastin treatment was sufficient to rescue the growth suppressive effects of both glutamine deprivation and CB-839 treatment in KPK cells with little effect on KP cells (*Figure 2C* and *Figure 2—figure supplement 1A*). Furthermore, through the examination of a panel of human LUAD cell lines, we demonstrate that *KEAP1* mutants are robustly sensitive to glutaminase inhibition (*Figure 2—figure supplement 1B*). Importantly, pre-treatment with Erastin overcomes sensitivity to glutaminase inhibition in *KEAP1* mutant lines (*Figure 2—figure supplement 1B*). xCT/Slc7a11 functions based on the intra- and extra- cellular gradients of cystine and glutamate in a concentration dependent manner (*Briggs et al., 2016 Watanabe and Bannai, 1987*). Therefore, we asked whether we could rescue glutamine dependency of KPK cells by forcing the export of cystine and the import of glutamate by modulating the concentration of these two amino acids in the culture media. By either raising the levels of glutamate or lowering the levels of cystine we were able to rescue proliferation in KPK cells after CB-839 treatment (*Figure 2D and E*). To further validate that CB-839 sensitivity in KPK cells is dependent on xCT/Slc7a11, we tested multiple doxycycline inducible mirE based shRNAs targeting Slc7a11 (*Figure 2—figure supplement 2A and B*). In line with expectations, knockdown of Slc7a11 by two independent shRNAs rescued KPK cells from both CB-839 treatment as well as glutamine deprivation while shCTRL had no effect (*Figure 2F* and *Figure 2—figure supplement 2C*). In addition, modulating the levels of glutamate or cystine in the media did not provide additional benefits to depletion of Slc7a11 in regards to sensitivity to CB-839 or glutamine

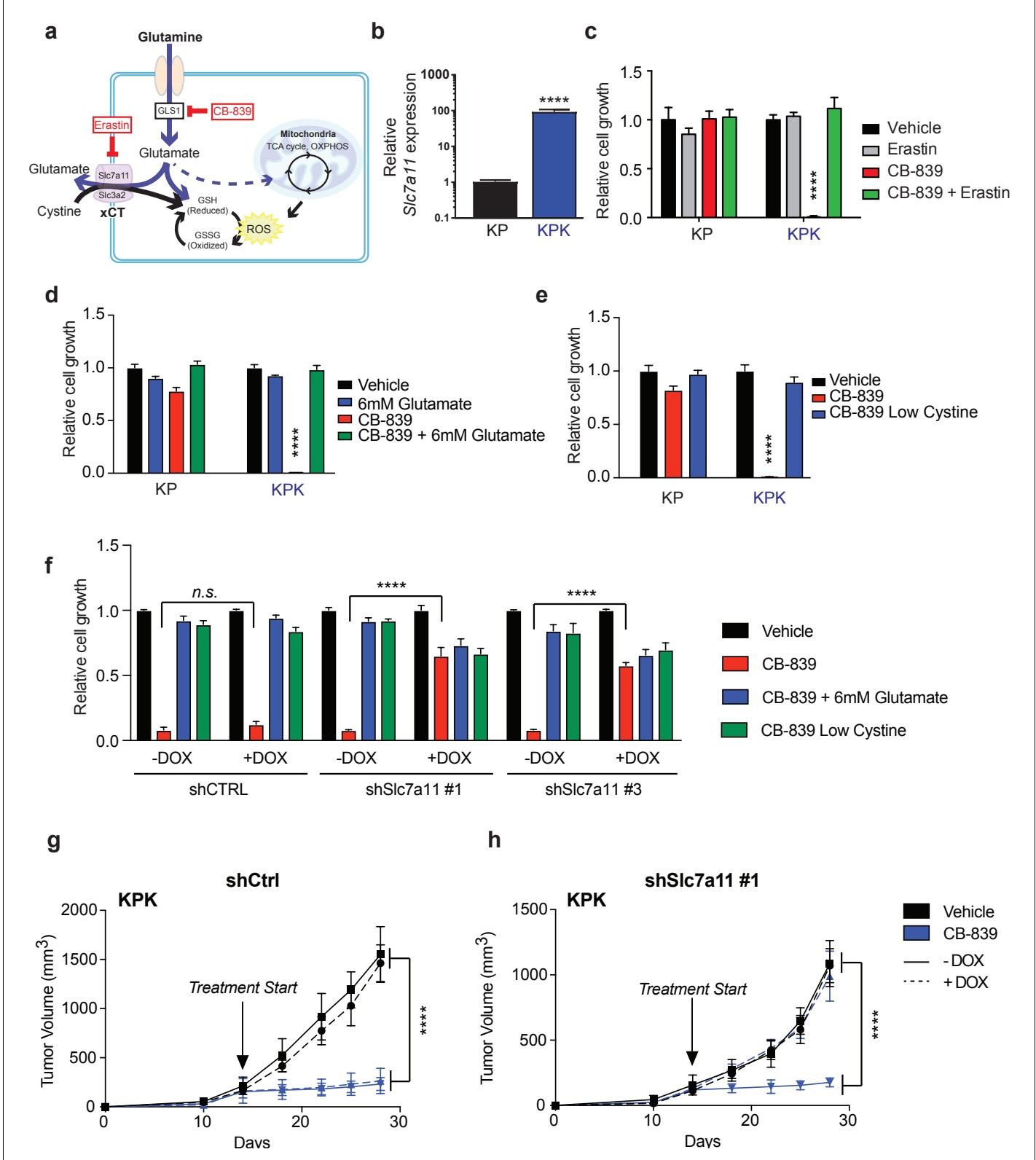

**Figure 2.** xCT/Slc7a11-dependent glutamate secretion in *Keap1* mutant cells causes glutamine dependency. (a) Schematic depicting glutaminolysis and subsequent glutamate export by the xCT/Slc7a11 antiporter. Treatment with the small molecule CB-839 inhibits Gls1, blocking the conversion of glutamine to glutamate while the small molecule Erastin inhibits Slc7a11 preventing glutamate export. (b) Quantitative real-time PCR of mRNA expression of *Slc7a11* in KP and KPK cells (*n* = 3, technical replicates). Data presented as relative to *Slc7a11* expression in KP cells. (c) Proliferation of KP

*Figure 2 continued on next page*

*Figure 2 continued*

and KPK cells after overnight pretreatment with 500 nM Erastin followed by 250 nM CB-839 treatment for 5 days (*n* = 3, triplicate wells). Data presented as relative to vehicle only treated condition for each cell line. (**d**) Proliferation of KP and KPK cells after addition of 6 mM glutamate followed by 250 nM CB-839 treatment for 5 days (*n* = 3, triplicate wells). Data presented as relative to vehicle only treated condition for each cell line. (**e**) Proliferation of KP and KPK cells in media containing low cystine (20 µM, 10X reduction from RPMI which contains 208 µM cystine) followed by 250 nM CB-839 treatment for 5 days (*n* = 3, triplicate wells). Data presented as relative to vehicle only treated condition for each cell line. (**f**) Proliferation of KP and KPK expressing either a doxycycline inducible control shRNA (shCTRL) or an shRNA targeted against Slc7a11 (shSlc7a11). Cells were cultured in RPMI with or without doxycycline that was supplemented with 6 mM glutamate or RPMI containing low cystine (20 µM, 10X reduction from RPMI which contains 208 µM cystine) followed by 250 nM CB-839 treatment for 5 days (*n* = 3, triplicate wells). (**g**) Subcutaneous tumor volumes of KPK tumors expressing a doxycycline inducible control shRNA (shCtrl). Animals were treated with vehicle (black) or CB-839 (blue) and received normal (solid line) or doxycycline (dashed line) feed starting from day 13 (arrow indicating treatment start, *n* = 6 tumors) (**h**) Subcutaneous tumor volumes of KPK tumors expressing a doxycycline inducible shRNA targeted against Slc7a11 (shSlc7a11). Animals were treated with vehicle (black) or CB839 (blue) and received normal (solid line) or doxycycline (dashed line) feed starting from day 13 (arrow indicating treatment start, *n* = 6 tumors). All error bars depict s.e.m. ****p<0.0001.

DOI: https://doi.org/10.7554/eLife.28083.004

The following figure supplements are available for figure 2:

**Figure supplement 1.** Inhibition of xCT/Slc7a11 rescues glutamine dependency in *Keap1* mutant cells.

DOI: https://doi.org/10.7554/eLife.28083.005

**Figure supplement 2.** Reduction in xCT/Slc7a11 expression rescues glutamine dependency both in vitro and in vivo.

DOI: https://doi.org/10.7554/eLife.28083.006

deprivation (*Figure 2F* and *Figure 2—figure supplement 2C*). Taken together, these results indicate that increased expression of xCT/Slc7a11 through Nrf2 hyperactivation in *Keap1* mutant cells is necessary for the increased glutamine dependency and sensitivity to glutaminase inhibitors.

*Keap1* mutations that activate Nrf2 result in elevated GSH production thereby leaving *Keap1* mutant cells highly dependent on glutaminolysis and susceptible to glutaminase inhibition in vitro. However, the extracellular environment can greatly affect cellular metabolism, and culturing cells in vitro versus in vivo can significantly alter their metabolic requirements (*Davidson et al., 2016*). To assess the importance of xCT/Slc7a11 in driving sensitivity to glutaminase inhibition in KPK cells in vivo, we implanted KP and KPK cells that expressed inducible shRNA constructs against Slc7a11 or CTRL subcutaneously in mice. After tumors were established, mice were randomized to groups to receive oral CB-839 or vehicle twice daily and to receive a doxycycline or control diet to induce shRNA expression. CB-839 treatment suppressed tumor growth only in KPK cells, but had no effect on KP tumors (*Figure 2G and H* and *Figure 2—figure supplement 2D and E*). Knockdown of Slc7a11 completely abrogated the effect of CB-839 on KPK tumors (*Figure 2H* and *Figure 2—figure supplement 2E*). These data strongly support the notion that *Keap1* mutations result in a metabolic liability to glutaminase inhibition in vivo driven by increased expression of xCT/Slc7a11 in KPK cells.

### *Keap1* mutants have defects in TCA cycle anaplerosis

*Keap1* mutant cells are dependent on glutamine due to basally decreased intracellular pools of glutamate, partially because of increased secretion via xCT and increased production of GSH. Glutamate can fuel both GSH synthesis, or via transamination be converted to α-KG, a carbon source for the TCA cycle (*Figure 3A*). To assess whether limited glutamate availability for GSH synthesis leads to proliferation defects, we pretreated KPK cells with the antioxidants trolox (Vitamin E analog) or N-acetyl cysteine (NAC). However, supplementation with these antioxidants did not rescue proliferation after glutamine deprivation or glutaminase inhibition in KPK cells (*Figure 3—figure supplement 1A* and *Figure 3B*). We hypothesized that high antioxidant production in KPK cells driven by Nrf2 activation increases cellular demand for glutamate. Glutamate is required in order to synthesize glutathione while limiting availability for other cellular processes such as fueling the TCA cycle. Consistent with this hypothesis, pretreatment with: (1) dimethyl 2-oxoglutarate (DMG), a cell permeable α-KG analogue, or (2) pyruvate, a glucose-derived TCA cycle carbon source, rescued sensitivity to CB-839 and glutamine deprivation in KPK cells (*Figure 3B* and *Figure 3—figure supplement 1A*). These results suggest that sensitivity to glutaminase inhibition is due to decreased glutamate availability for anaplerosis or other biosynthetic reactions, not synthesis of the antioxidant GSH. In line with this, reducing GSH levels in KPK cells with L-Buthionine sulfoximine (BSO) did not alter sensitivity to CB-839 (*Figure 3—figure supplement 1B and C*).

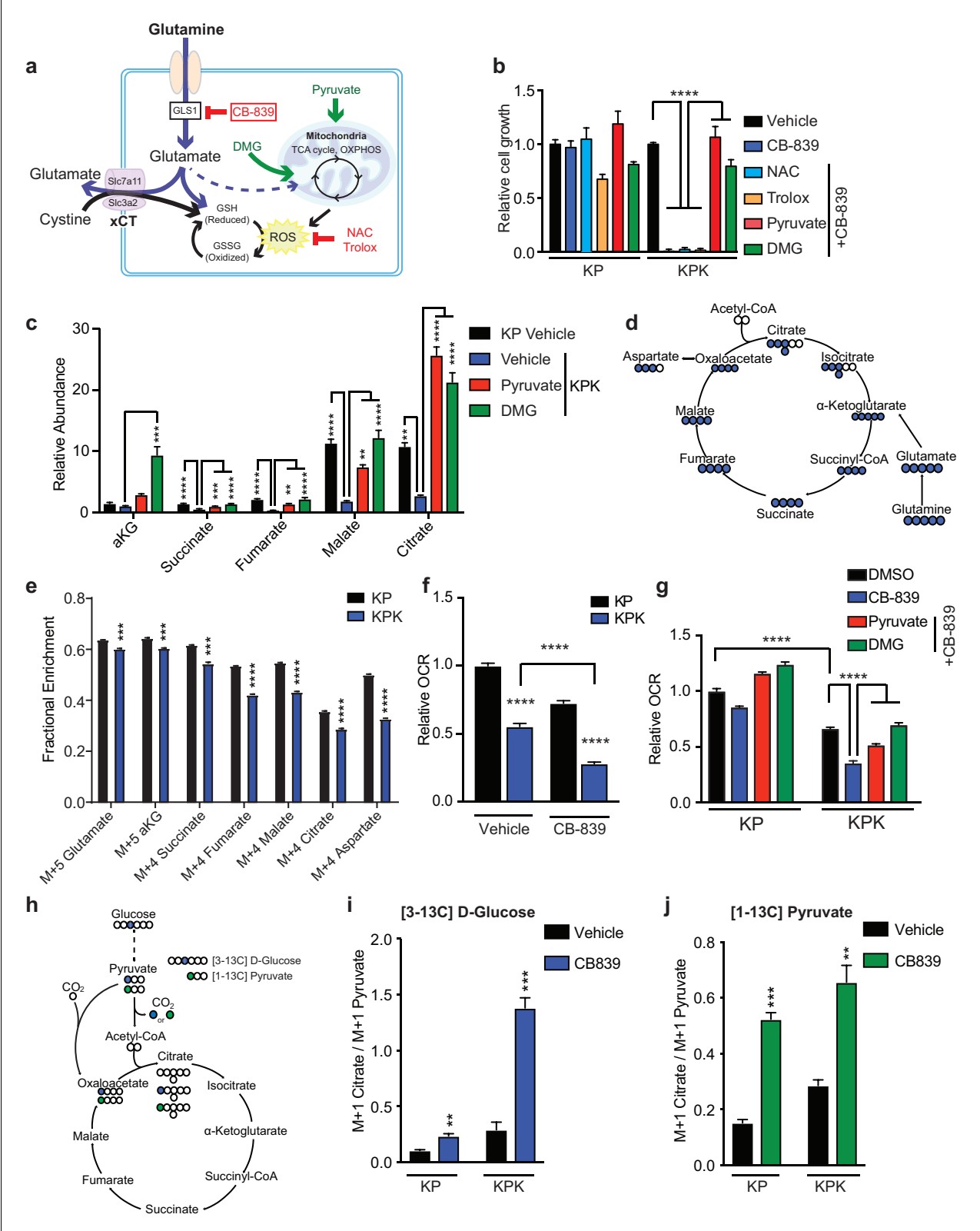

**Figure 3.** *Keap1* mutants have defects in glutamine anaplerosis. (a) Schematic depicting glutaminolysis and subsequent glutamate export by the xCT/ Slc7a11 antiporter. Treatment with the small molecule CB-839 inhibits Gls1, blocking the conversion of glutamine to glutamate. Treatment with the antioxidants N-acetylcysteine (NAC) or trolox scavenge cellular reactive oxygen species (ROS) while treatment with either pyruvate or dimethyl 2-oxoglutarate (DMG) provide carbons to the TCA cycle (b) Proliferation of KP and KPK cells after overnight pretreatment with either 500 nM NAC, 500

*Figure 3 continued on next page*

*Figure 3 continued*

nM Trolox, 2 mM pyruvate or 2 mM DMG followed by 250 nM CB-839 treatment for 5 days ($n = 3$, triplicate wells). Data presented as relative to vehicle only treated condition for each cell line. (c) Relative abundance of TCA cycle metabolites in KP and KPK cells supplemented with 2 mM pyruvate or DMG where indicated ($n = 3$, triplicate wells). Data is normalized by cell counts for each cell line in each condition. (d) Schematic depicting the TCA cycle. Filled blue circles represent $^{13}C$ atoms derived from [U-$^{13}C$]-L glutamine. (e) Mass isotopomer analysis of TCA cycle metabolites in KP and KPK cells cultured for 8 hr with [U-$^{13}C$]-L glutamine ($n = 3$, triplicate wells). (f) Relative mitochondrial respiration measured by oxygen consumption rate (OCR, pmol $O_2$/min) in KP and KPK cells treated either with vehicle or 250 nM CB-839 for 4 hr ($n = 5$, replicate wells). Data is presented as relative to vehicle treated KP cells. (g) Relative mitochondrial respiration measured by oxygen consumption rate (OCR, pmol $O_2$/min) in KP and KPK cells pretreated either 2 mM pyruvate or 2 mM DMG followed by treatment with vehicle or 250 nM CB-839 for 4 hr ($n = 3$, triplicate wells). Data is presented as relative to vehicle treated KP cells. (h) Schematic depicting the entry of glucose derived carbon into the TCA cycle via pyruvate carboxylase activity. Filled circles represent $^{13}C$ atoms derived from [3$^{13}C$]-D glucose (blue) or [1$^{13}C$]-pyruvate (green). Carbon flux through pyruvate carboxylase results in M + 1 labeled citrate while flux through pyruvate dehydrogenase results in unlabeled citrate. (i) Mass isotopermer analysis of citrate in KP and KPK cells cultured for 8 hr with [3$^{13}C$]-D glucose and 250 nM CB-839 where indicated ($n = 3$, triplicate wells). Data is normalized with respect to M + 1 pyruvate to account for differences in basal rates of glycolysis. (j) Mass isotopermer analysis of citrate in KP and KPK cells pretreated with [1$^{13}C$]-pyruvate overnight and then treated 250 nM CB-839 for 8 hr where indicated ($n = 3$, triplicate wells). Data is normalized with respect to M + 1 pyruvate to account for differences in basal rates of glycolysis. All error bars depict s.e.m. *$p<0.05$, **$p<0.01$, ****$p<0.0001$.

DOI: https://doi.org/10.7554/eLife.28083.007

The following figure supplements are available for figure 3:

**Figure supplement 1.** Antioxidants do not rescue glutamine dependency in *Keap1* mutant cells.
DOI: https://doi.org/10.7554/eLife.28083.008
**Figure supplement 2.** Fractional labeling of glutamine derived carbon in TCA cycle intermediates.
DOI: https://doi.org/10.7554/eLife.28083.009
**Figure supplement 3.** Basal respiration in KP and KPK cells.
DOI: https://doi.org/10.7554/eLife.28083.010

Our data suggests that hyperactivation of Nrf2 by *Keap1* mutation genetically reprograms cells to produce high levels of antioxidants at the expense of contributing carbon to replenish the TCA cycle. Consistent with this data, KPK cells have reduced total levels of TCA cycle intermediates (*Figure 3C*) and supplementation with either 2 mM pyruvate or DMG increased total levels of TCA cycle intermediates in KPK cells equivalent to or in excess to those in KP cells (*Figure 3C*). To determine the fate of glutamine, we used a stable-isotope labeled glutamine tracer (U-C$^{13}$-L-glutamine) (*Figure 3D*). We observed that KPK cells incorporate less carbon from glutamine into the TCA cycle as compared to KP cells (*Figure 3E*). Taken together, these data strongly suggest that KPK cells have a basal deficiency in glutamine metabolism that contributes to their increased sensitivity to loss of glutamine-derived glutamate for anaplerosis. A functional output of the TCA cycle is oxidative phosphorylation in the mitochondria, as the TCA cycle generates NADH and $FADH_2$ that will be used by the mitochondrial electron transport system to produce ATP. Measuring mitochondrial respiration revealed that KPK cells have reduced oxygen consumption rates (OCR) compared to KP cells indicating that *Keap1* mutants have lower mitochondrial respiration basally (*Figure 3F*). The overall decrease in respiration in KPK compared to KP cells is in line with decreased TCA cycle output (*Figure 3C*). However, supplementation with pyruvate or DMG did not significantly increase basal OCR in KPK cells (*Figure 3—figure supplement 3A*). This is likely a result of KPK cells having reduced reserve respiratory capacity as compared to KP cells (*Figure 3—figure supplement 3B*) in normal media conditions. As expected, CB-839 treatment reduced respiration in both KP and KPK cells, however, the drop in OCR after CB-839 administration was dramatically more pronounced in KPK cells (−50.8%) compared to KP cells (−21.6%), despite baseline levels being lower in KPK cells (*Figure 3F and G*). The drop in OCR in response to CB-839 was rescued by co-treatment with DMG or pyruvate (*Figure 3G*). The ability of both DMG and pyruvate to rescue overall levels of TCA cycle metabolites as well as cellular proliferation and mitochondrial respiration in response to CB-839 treatment in KPK cells suggests that the basal defects in TCA cycle metabolism in KPK cells causes a metabolic bottleneck due to limited glutamate availability which is further exacerbated by glutaminase inhibition via CB-839. As an αKG analogue, DMG can directly fuel the TCA cycle and bypass glutaminase inhibition. However, the ability of pyruvate to rescue TCA cycle defects in KPK cells is less direct. One explanation is that pyruvate offers an alternative anaplerotic substrate via increased flux through pyruvate carboxylase, which catalyzes the conversion of pyruvate to oxaloacetate. To determine if treatment with CB-839 enhanced carbon flux through pyruvate carboxylase (PC), we

performed stable isotope tracing using [3 C$^{13}$]-D-glucose to look at PC flux in basal conditions as well as with [1 C$^{13}$]-pyruvate to look at flux in rescue conditions (*Figure 3H*). In line with this hypothesis, we observed increased flux through pyruvate carboxylase in response to CB-839 in normal media conditions as well as when cells are supplemented with pyruvate (*Figure 3I and J*). This is in agreement with previous results indicating that KP tumors have increased PC flux (*Davidson et al., 2016*).

Taken together these data show that *Keap1* mutations result in basal defects in central carbon metabolism, with KPK cells having overall decreased levels of TCA cycle intermediates and reduced contribution of carbon from glutamine to these metabolites. This metabolic imbalance in KPK cells can be therapeutically exploited by the use of glutaminase inhibitors to further limit glutamate availability.

## Nrf2 activation is sufficient to sensitize cells to glutaminase inhibition and *Keap1* mutations predict sensitivity across multiple cancer types

Having established that *Keap1* mutations increase sensitivity to glutaminase inhibition, we sought to determine if activation of Nrf2 in wild type cells would be sufficient to sensitize cells to CB-839. Using a small-molecule activator that disrupts the binding of Keap1 to Nrf2, KI696 (*Davies et al., 2016*), we first validated that treatment with KI696 could increase Nrf2 protein levels as well as increase expression of downstream target genes in KP cells (*Figure 4—figure supplement 1A–C* and *Figure 4* 1A). Pretreatment with KI696 and subsequent Nrf2 activation was sufficient to induce strong growth suppression in response to CB-839 treatment in KP cells (*Figure 4B*), similar to what we observed in KPK cells (*Figure 2C*). Additionally, effects of CB-839 were rescued both by pretreatment of anaplerotic precursors, pyruvate and DMG, and by modulating extracellular glutamate and cystine concentrations, but not by the antioxidants NAC and trolox (*Figure 4B*).

Additionally, we took a genetic approach to elevate GSH levels in KP cells in order to reproduce the *Keap1* mutant phenotype in wild type cells and sensitize them to CB-839. Using the CRISPR synergistic activation mediate (CRISPRa SAM) system (*Konermann et al., 2015*) (*Figure 4C*), which takes advantage of the DNA targeting capabilities of a catalytically dead Cas9 protein fused to VP64 and a modified sgRNA that recruits and targets transcriptional activators to promoter regions upstream of transcriptional start sites of genes of interest to enhance endogenous transcription. Using sgRNAs that target the promoters of *Gclc, Gclm,* and *Slc7a11*, we were able to increase mRNA expression in cells that expressed single sgRNAs to each gene as well as in cells that expressed two sgRNAs to increase expression of *Slc7a11* and *Gclc* or *Gclm* in combination (*Figure 4C*). Increased expression of these Nrf2 target genes resulted in significant increases in total GSH in all cell lines (*Figure 4—figure supplement 1D*). However, we only observed increased sensitivity to CB-839 in KP cells that had increased expression of *Slc7a11* in combination with either *Gclc* or *Gclm* (*Figure 4D*).

Similar to mouse KP cells, human *KEAP1* wild type lung cancer cells, H2009, responded to KI696 with nuclear accumulation of NRF2 and subsequent up regulation of target gene expression and total GSH levels (*Figure 4E* and *Figure 4—figure supplement 2A–C*). In line with KP cells, human H2009 cells responded strongly to co-treatment with CB-839 and KI696 and this synergistic effect was rescued by pretreatment of anaplerotic precursors, modulating extracellular glutamate and cystine concentrations but not by antioxidants (*Figure 4F*). We next asked if KI696 treatment would replicate the metabolic phenotypes observed in KPK cells. KP cells treated with KI696 had reduced total levels of TCA cycle intermediates (*Figure 4G*), equivalent to that observed in KPK cells (*Figure 4G* and *Figure 3C*). Additionally, overall mitochondrial respiration was reduced upon treatment of wild type KP cells with KI696, and further reduced by the addition of CB-839 (*Figure 4H*). These data demonstrate that Nrf2 activation is sufficient to rewire cellular metabolism, resulting in increased glutamine dependency and sensitivity to CB-839 treatment in both mouse and human LUAD lines.

In addition to LUAD, *KEAP1/NRF2* mutations are found in other cancers (*Shibata et al., 2008a*; *Kim et al., 2010*; *Konstantinopoulos et al., 2011*; *Muscarella et al., 2011*; *Cancer Genome Atlas Research Network, 2014*; *Hanada et al., 2012*; *Jaramillo and Zhang, 2013*; *Sato et al., 2013*; *Goldstein et al., 2016*; *Fabrizio et al., 2017*). To assess whether NRF2 promotes glutamine dependency in other *KEAP1* mutant cancers, we tested CB-839 on a panel of human cancer cell lines with WT or mutant *KEAP1*, including melanoma, bone, colon, renal, squamous and urinary tract cancers.

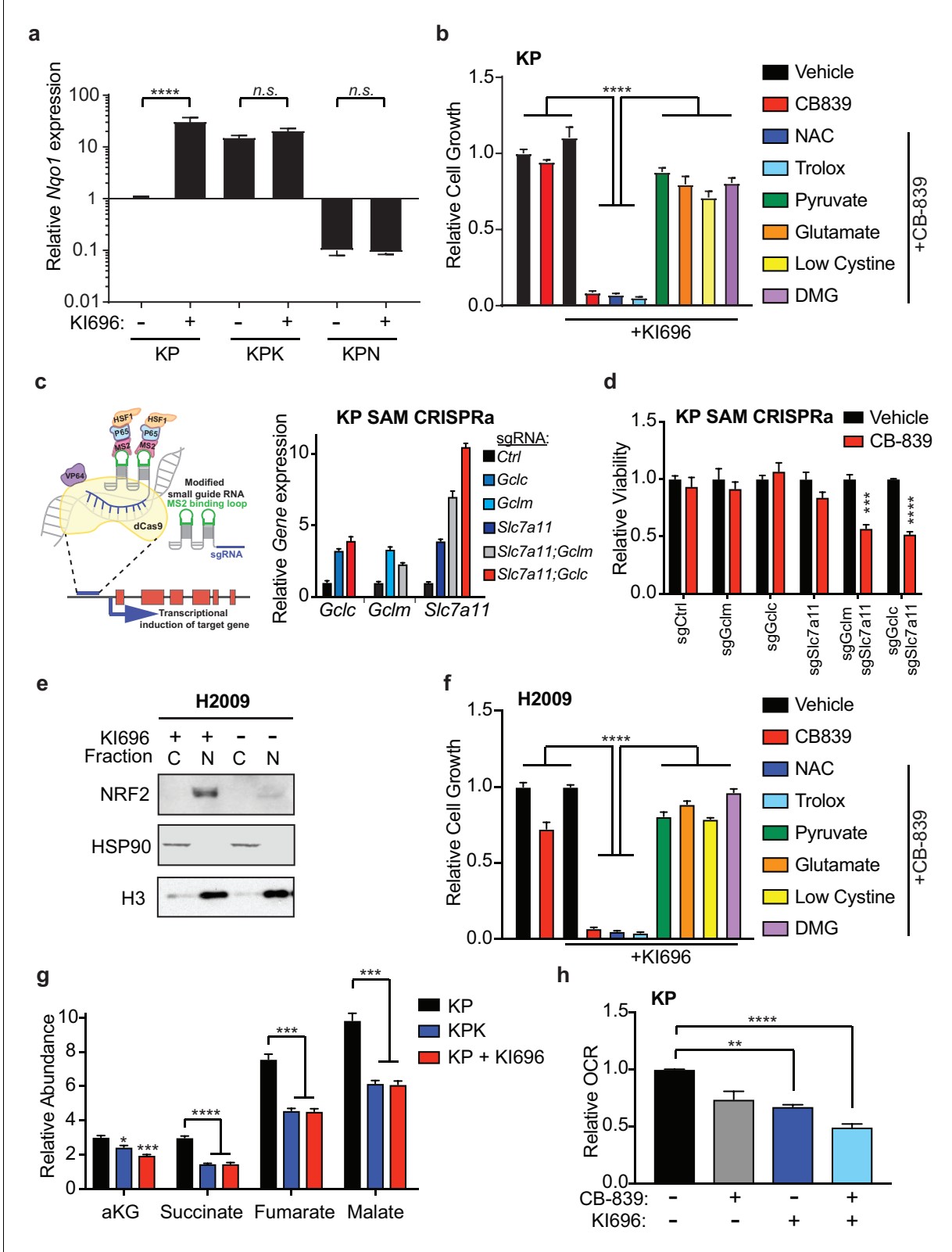

**Figure 4.** Nrf2 activation is sufficient to sensitize cells to glutaminase inhibition. (a) Quantitative real-time PCR of mRNA expression of *Nqo1* in KP, KPK, and KPN (Nrf2 null) cells after pretreatment with 1 µM KI696 for 36 hr (n = 3, technical replicates). Data presented as relative to *Nqo1* expression in untreated KP cells. (b) Proliferation of KP cells after pretreatment with the small molecule activator of Nrf2, 1 µM KI696, followed by treatment with either 500 nM NAC, 500 nM trolox, 2 mM pyruvate, 2 mM DMG, supplementation with 6 mM glutamate, or media containing low cystine (20 µM, 10X
*Figure 4 continued on next page*

*Figure 4 continued*

reduction from RPMI which contains 208 µM cystine) followed by 250 nM CB-839 treatment for 5 days (*n* = 5, replicate wells). Data is presented as relative to vehicle treated. (**c**) **Left:** Schematic depicting CRISPR synergistic activation mediated (SAM) system. A modified sgRNA recruits catalytically dead Cas9 that is fused to VP64 to promoter regions up stream of the transcription start site of target genes and recruits additional transcriptional activators to induced endogenous transcription of target genes. **Right:** Quantitative real-time PCR of mRNA expression of *Gclc, Gclm,* and *Slc7a11* in KP SAM CRISPRa cell lines with sgRNAs targeting each gene of interest (*n* = 3, technical replicates). Data presented as relative to expression of each gene in sgCtrl cells. (**d**) Proliferation of KP SAM CRISPRa cells expressing sgRNAs against Gclc, Gclm, and Slc7a11. Cells were seeded in media lacking glutamate and after 24 hr were treated with 100 nM of CB-839 for 5 days (*n* = 3, replicate wells). Data is presented as relative to vehicle treated control for each cell line. (**e**) Western blot depicting Nrf2 expression in *KEAP1* wild type human lung adenocarcinoma line (H2009) after treatment with KI696. Cells were treated with 1 µM KI696 for 36 hr and then nuclear (**N**) and cytoplasmic (**C**) fractions were collected. First panel depicts an Nrf2 blot, second panel depicts an Hsp90 loading control for cytoplasmic fraction, and third panel depicts an H3 loading control for nuclear fraction. **f**) Proliferation of *KEAP1* wild type human lung cancer cell line H2009 after pretreatment with the small molecule activator of NRF2, 1 µM KI696, followed by treatment with either 500 nM NAC, 500 nM trolox, 2 mM pyruvate, 2 mM DMG, supplementation with 6 mM glutamate, or media with reduced cystine (20 uM) followed by 250 nM CB-839 treatment for 5 days (*n* = 5, replicate wells). Data is presented as relative to vehicle treated. (**g**) Relative abundance of TCA cycle metabolites in KP, KPK, and KP cells treated with 1 µM KI696 (*n* = 3, triplicate wells). Data is normalized by cell counts for each cell line. (**h**) Relative mitochondrial respiration measured by oxygen consumption rate (OCR, pmol $O_2$/min) in KP cells treated with either vehicle or 1 µM KI696 followed by treatment with vehicle or 250 nM CB-839 for 4 hr (*n* = 3, triplicate wells). Data is presented as relative to vehicle treated KP cells. All error bars depict s.e.m. *$p<0.05$, **$p<0.01$, ***$p<0.001$, ****$p<0.0001$.

DOI: https://doi.org/10.7554/eLife.28083.011

The following figure supplements are available for figure 4:

**Figure supplement 1.** Small molecule KI696 induces Nrf2 activation in KP cells.
DOI: https://doi.org/10.7554/eLife.28083.012

**Figure supplement 2.** Small molecule KI696 induces Nrf2 activation in *KEAP1* wild type human lung cancer cells.
DOI: https://doi.org/10.7554/eLife.28083.013

For each cancer type, mutations in *KEAP1* predicted sensitivity to CB-839 (*Figure 5A*). Importantly, co-treatment with KI696 was able to sensitize *KEAP1* wild type cancer cells that were relatively insensitive to CB-839 (*Figure 5B* and *Figure 5—figure supplement 1*). Taken together, our data suggests that glutaminase is an attractive target in both lung adenocarcinoma and in other cancers in which the KEAP1/NRF2 axis is mutated and the glutaminase inhibitor CB-839 could be an effective therapeutic agent in patients with these genetic subtypes of cancer. Additionally, glutaminase inhibition could be more broadly applicable in a variety of cancer types when combined with a small molecule activator of NRF2, which renders *KEAP*1 wild type cells highly sensitive to glutaminase inhibition.

## Discussion

Tumorigenesis requires cancer cells to increase their metabolic output to support tumor growth. Higher metabolic activity increases oxidative stress in the form of ROS, which can lead to oxidative damage of macromolecules. Under physiologic conditions of increased ROS, the NRF2 stress response promotes an antioxidant network that will scavenge and clear ROS. After ROS clearance NRF2 will once again be degraded through its interaction with KEAP1 and CUL3-mediated ubiquitination. However, in the context of chronic ROS exposure, post-transcriptional or post-translational signaling which stabilize NRF2 (*Goldstein et al., 2016*), or epigenetic (*Muscarella et al., 2011*; *Hanada et al., 2012*; *Fabrizio et al., 2017*) and genetic events that lead to NRF2 stabilization (*Shibata et al., 2008a*; *Kim et al., 2010*; *Konstantinopoulos et al., 2011*; *Jaramillo and Zhang, 2013*; *Sato et al., 2013*), NRF2 and its downstream antioxidant program will be chronically activated (*Figure 5C*). NRF2 activation subsequently leads to changes in cellular metabolism to support increased synthesis of antioxidants (*Mitsuishi et al., 2012*; *DeNicola et al., 2015*), which have recently been shown to promote tumorigenesis (*Romero et al., 2017*). In this study, we show that NRF2 dependent metabolic alterations result in a unique set of metabolic requirements in *KEAP1* mutant cells that can consequently be theraputically exploited.

We show that constitutive (*KEAP1* mutation) or pharmacological activation of NRF2 leads to alterations in glutamine metabolism and defective glutamate-dependent fueling of central carbon metabolism (*Figure 5D*). Glutamine is a critical nutrient for proliferating cells as it contributes to supporting cellular bioenergetics by fueling ATP production as well as supplying the cell with necessary intermediates for biosynthesis (*Deberardinis et al., 2008*). Both the carbon and nitrogen

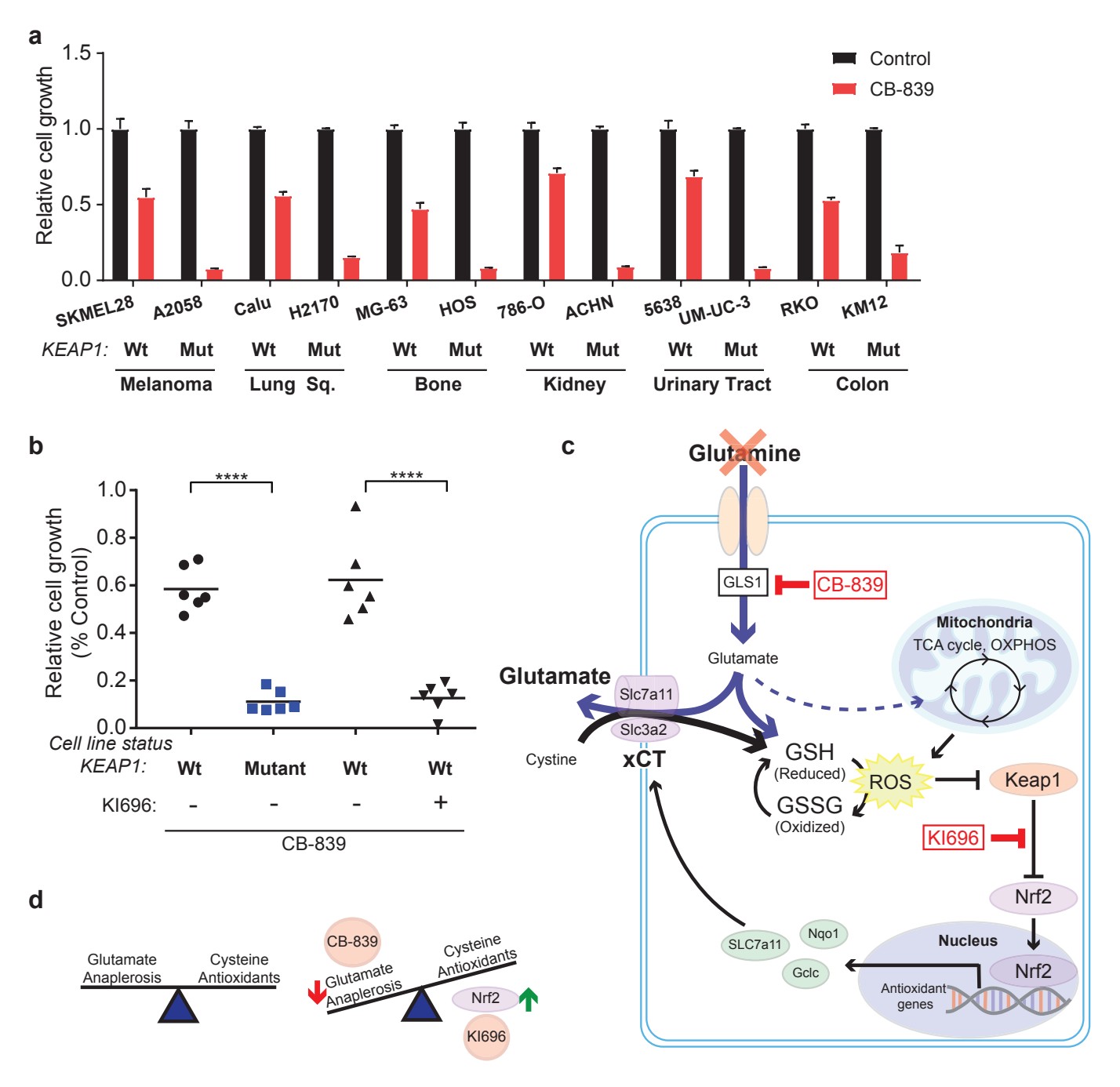

**Figure 5.** *KEAP1* mutations predict sensitivity to glutaminase inhibitors across multiple cancer subtypes. (a) Proliferation of a panel of human cancer cell lines of various tissues of origin treated with vehicle or 250 nM CB-839 for 4 days. One wild type and one *KEAP1* mutant cell line assessed for each tissue (*n* = 3, triplicate wells). Data is presented as relative to vehicle treated. (b) Proliferation of cell lines shown in (a) along with pretreatment of 1 µM KI696 in *KEAP1* wild type cells followed by 250 nM CB-839 treatment for 4 days. Each data point represents an independent cell line (*n* = 3, triplicate wells). Data is presented as relative to vehicle treated. (c) Schematic depicting how constitutive NRF2 activation via *KEAP1* mutations or use of small molecule activator, KI696, results in increased glutathione production and high xCT/SLC7A11 expression. Increased GSH production increases cellular demand for glutamate, making *KEAP1* mutant cells glutamine dependent and susceptible to glutaminase inhibition. (d) Schematic depicting the balance between cellular production of antioxidants and demand for cysteine versus using glutamate to fuel anaplerosis (left). Activation of Nrf2 results in an imbalance between production of antioxidants and use of glutamate for anaplerosis that can be exploited by CB-839 treatment to further limit glutamate availability (right). All error bars depict s.e.m. ****$p<0.0001$.

DOI: https://doi.org/10.7554/eLife.28083.014

*Figure 5 continued on next page*

*Figure 5 continued*

The following figure supplement is available for figure 5:

**Figure supplement 1.** Activation of NRF2 sensitizes *KEAP1* wild type cells to glutaminase inhibition in multiple cancer types.
DOI: https://doi.org/10.7554/eLife.28083.015

derived from glutamine are used for the synthesis of nucleotides, amino acids, and other macromolecules. Nitrogen from glutamine contributes to de novo nucleotide synthesis making an adequate supply of glutamine to support purine and pyrimidine synthesis necessary to maintain rapid rates of proliferation (*Tardito et al., 2015*). Additionally, carbon and nitrogen from both glutamine and glutamate support amino acid synthesis via the activity of a variety of transaminases and aminotransferases (*DeBerardinis and Cheng, 2010*). Depleted intracellular glutamate levels in *KEAP1* mutant cells likely alters the activity of various transaminases due to limited substrate availability and may also result in the dependency on other exogenously supplied metabolites (*DeBerardinis and Chandel, 2016*). The decreased availability of intracellular glutamate we observe in *KEAP1* mutant cells results in dependency on exogenous glutamine (*Figure 1—figure supplement 1F*), which can be therapeutically exploited by glutaminase inhibition (*Figures 1–5*).

The dependency on glutamine-derived glutamate that we observe in cells with NRF2 activation is primarily due to defective TCA cycle anaplerosis and not the limited availability of glutamate for antioxidant production, since neither antioxidant supplementation (*Figure 3B*, *Figure 3—figure supplement 1A*, and *Figure 4A*) nor pharmacological decrease of GSH levels (*Figure 3—figure supplement 1B and C*) rescued glutamine dependency or CB-839 sensitivity. The ability of both pyruvate and DMG to rescue overall levels in TCA cycle metabolites as well as cellular proliferation and reduced mitochondrial respiration in response to CB-839 treatment highlights that glutaminase inhibition further depletes limited intracellular glutamate stores in *KEAP1* mutant cells and exacerbates basal defects in the TCA cycle. Reduced TCA cycle output in *KEAP1* mutant cells can have wide ranging effects beyond immediate use in biosynthesis pathways, it may also impact the activity of αKG-dependent deoxygenases, a family of enzymes that regulate signaling, metabolism, and epigenetic modifications (*Loenarz and Schofield, 2008*). αKG, succinate, and fumarate levels can have wide ranging effects beyond immediate use in biosynthesis pathways by altering hypoxia-inducible transcription factor (HIF) signaling (*Selak et al., 2005*; *King et al., 2006*; *Boulahbel et al., 2009*) and altering gene expression through epigenetic modifications (*Xiao et al., 2012*; *Letouzé et al., 2013*; *Waterfall et al., 2014*).

We show that the decrease in intracellular glutamate availability for the TCA cycle, in the context of increased NRF2, occurs through the active secretion of glutamate through xCT/SLC7A11. We hypothesize that xCT/SLC7A11-mediated secretion of glutamate and import of cystine is driven by the increased intracellular demand for cysteine, a key amino acid for many NRF2-induced antioxidants (e.g. GSH, TXN). Alternatively, the levels of glutamate and cystine in culture medium in vitro or in serum in vivo can also modulate the directionality of xCT/SLC7A11 (*Briggs et al., 2016*), which may also change the availability of intracellular glutamate for central carbon metabolism and the dependency on exogenous glutamine (Muir, A. *et al*, co-submitted with this manuscript). Similar to our findings, depletion of xCT/SLC7A11 improves cancer cell viability after glucose withdrawal through conservation of intracellular glutamate levels to maintain mitochondrial respiration (*Koppula et al., 2017*; *Shin et al., 2017*). In this context, *SLC7A11* expression was also dependent on the transcription factor ATF4, which is known to cross talk with NRF2 in cancer cell metabolism (*DeNicola et al., 2015*).

NRF2 is a pleiotropic transcription factor that regulates many genes and pathways that could contribute to the depletion of intracellular glutamate stores in *Keap1* mutants and facilitate metabolic rewiring (*Taguchi et al., 2011*). In line with this, using pharmacological and genetic means to modulate GSH levels specifically was not sufficient to alter sensitivity to CB-839 (*Figure 3—figure supplement 1D* and *Figure 4D*). Interestingly, increased endogenous expression of *Slc7a11* in combination with *Gclc* or *Gclm* using SAM CRISPRa (*Sayin and Papagiannakopoulos, 2017*), was sufficient to induce sensitivity to CB-839 in KP SAM CRISPRa cells (*Figure 4D*). However, the sensitivity to CB-839 was not as robust as in KPK cells, suggesting that the moderate increase in target genes and

GSH synthesis in double sg KP SAM CRISPRa cells was not enough to fully recapitulate the metabolic phenotype of *Keap1* mutant cells.

Tumor progression occurs in a complex microenvironment with infiltrating immune cells and fibroblasts, which can impact tumor growth (*Sleeman, 2012*). In addition to its cell-autonomous effects, glutamate secretion can signal to cells in the microenvironment through non cell-autonomous mechanisms (*Briggs et al., 2016*). Increased glutamate secretion and paracrine signaling from tumors with NRF2 activation may have a profound impact on both the function and the cellular composition of cells in the microenvironment (*Romero et al., 2017*). Altered levels of both glutamine and glutamate in the tumor microenvironment of *KEAP1* mutant tumors can effect cell signaling and regulate cell growth and proliferation pathways (*Rhoads et al., 1997*; *Larson et al., 2007*; *Nicklin et al., 2009*).

Glutamine metabolism is positively regulated by a variety of oncogenes, rendering many cancer types addicted to glutamine and making glutamine metabolism an attractive therapeutic target (*Wise and Thompson, 2010*). Hence, there are several small molecule glutaminase inhibitors, one of which, CB-839, is currently in phase I clinical trials (*Altman et al., 2016*). However, our data as well as previous studies highlight the potential importance of stratifying patients based on genotype for a successful response to glutaminase inhibitors. In a previous study, we have reported that *KEAP1* mutant tumors respond to CB-839 treatment, while growth of wild type tumors is unaffected in vivo (*Romero et al., 2017*). Taken together with the current study, which provides mechanistic insight into how *KEAP1* mutations rewire tumor metabolism and why glutamine deprivation results in growth suppression, these data provide critical pre-clinical evidence supporting the treatment of patients with *KEAP1* mutations with glutaminase inhibitors. We further provide evidence that mutations in *KEAP1* are sufficient to predict sensitivity to glutaminase inhibitors regardless of tissue of origin. CB-839 treatment significantly suppressed cell growth in a panel of tumor cells including melanoma, bone, colon, renal, squamous, and urinary tract cancers with *KEAP1* mutations. Additionally, use of KI696, a small molecule activator of NRF2, which blocks the interaction between NRF2 and KEAP1, was able to sensitize both mouse and human *KEAP1* WT cell lines to glutaminase inhibition. These data argue that the use of glutaminase inhibitors along with small molecule activators of NRF2 could be an effective therapeutic strategy with broad application across various *KEAP1* mutant and wild-type cancers.

## Materials and methods

### Cell lines and culture

KP and KPK isogenic clonal cell lines were previously established (*Romero et al., 2017*). Human cell lines were acquired from ATCC. All lines were tested negative for mycoplasma. All cells were cultured in a humidified incubator at 37°C and 5% $CO_2$. Cells were maintained in either DMEM or RPMI-1640 (Cellgro, Corning; Manassas, Virginia) supplemented with 10% fetal bovine serum (Sigma Aldrich, St. Louis, Missouri) and gentamicin (Invitrogen, ThermoFisher; Waltham, Massachusetts).

### Cell proliferation and viability assays

For cell proliferation assays conducted under different drug or media conditions as indicated, cells growing in DMEM were trypsinized, counted and plated into 12 well plate dishes (BD/Falcon, Corning; Manassas, Virginia) in 1 mL of RPMI media. For glutamine deprivation experiments, cells were seeded in RPMI containing either 2 mM or 0.5 mM glutamine at the time of plating. After cell attachment, cells were treated with the indicated drugs: vehicle (DMSO), 500 nM erastin (Sigma Aldrich) 6 mM glutamate (Sigma Aldrich), 50 uM Trolox (Acros Organics, Fisher Scientific; Waltham, Massachusetts), 0.5 mM *N*-acetyl-L-cysteine (NAC, Sigma Aldrich), 2 mM pyruvate (Gibco, ThermoFisher Scientific; Waltham, Massachusetts), 2 mM dimethyl-2-oxoglutarate (DMG, Sigma Aldrich), L-Buthionine sulfoximine (Sigma Aldrich), hydrogen peroxide (LabChem, Zelienople, Pennsylvania), menadione (Sigma Aldrich) or 1 μM KI696 (provided by Craig Thomas). Cells exposed to galactose oxidase (Sigma Aldrich) were cultured in media supplemented with 5 mM galactose (Sigma Aldrich). After 12 hr, cells were treated with 250 nM CB839. For media in which cystine concentrations were lowered to 20 μM, RPMI was prepared from a powder mix without amino acids (US Biological, Salem, Massachusetts) according to manufacturers instructions. All amino acids besides cystine were added to the

RPMI mixture in the same concentrations as present in RPMI-1640 formulation. Proliferation experiments were carried out for 5 days post drug treatment and collected either by staining or cell counting. Cell counts were collected by trypan blue exclusion on a Countess II automated cell counter (Life Technologies, ThermoFisher Scientific; Waltham, Massachusetts). Cells were stained with a 0.5% crystal violet solution in 25% methanol. Plates were then washed, dried, and crystal violet was eluted in 400 uL of 10% acetic acid.

For cell viability assays cells were plated in a white, opaque 96-well plate with clear bottom at a density of 1000 cells/well in RPMI. Drugs were added at the same concentration and manner as described above for the indicated amount of time. After 4 days, cell viability in the presence of all compounds was assessed by cell titer glo (Promega #G7570, Madison, Wisconsin).

## qPCR analysis

mRNA was collected from cells with RNeasymini kit (Qiagen, Germany) according to manufacturers protocol. Complementary DNA (cDNA) was synthesized from extracted mRNA using the High Capacity cDNA Reverse Transcription Kit (Applied Biosystems #4368814, ThermoFisher Scientific; Waltham, Massachusetts) according to manufacturers protocol. Gene expression was analyzed by quantitative reverse transcription polymerase chain reaction on a CFX96 Real-Time System (BioRad Technologies, Hercules, California) with the following primers, *Slc7a11*: gattcatgtccacaagcacac and gagcatcaccatcgtcagag, *SLC7A11*: ccatgaacggtggtgtgtt, *Gclc*: agatgatagaacacgggaggag and tgatcctaaagcgattgttcttc, *GCLC*: atgccatgggatttggaat and gatcataaaggtatctggcctca, *Nqo1*: acgctgccatgtatgacaaa and ggatcccttgcagagagtaca, *NQO1*: acgctgccatgtatgacaaa and ggatcccttgcagagagtaca, *Gclm:* tgactcacaatgacccgaaa and tcaatgtcagggatgctttct.

## Immunoblotting

Cells were lysed in 150 uL of ice-cold RIPA buffer supplemented with 1X Complete Mini inhibitor mixture (Roche, #11 836 153 001, Switzerland) and mixed on a rotator at 4°C. Protein concentration of lysates was determined using the Bio-Rad DC Protein Assay (#500–0114). 20–40 ug of total protein was separated on a 4–12% Bis-Tris gradient gel (Bio-Rad) by SDS-PAGE and then transferred to nitrocellulose membrane (Bio-Rad). The following antibodies were used for immunoblotting: anti-Nrf2 (Cell Signaling Technologies, Danvers, Massachusetts, #12721, 1:1000), anti-HSP90 (BD, Sparks, Maryland, #610418, 1:4000), anti-H3 (AbCam, Cambridge, Massachusetts, #1991, 1:4000), and anti-Slc7a11 (Novus Biologicals, Littleton, Colorado, #NB300-318).

## Glutathione assay

Total glutathione was measured after treatment with 250 nM CB-839 or treatment with BSO for 24 hr where indicated and collected according to manufacturer's protocol (Promega, V6611). Data was normalized to additional wells that received identical conditions using Cell Titer Glo (Promega, G7570).

## ROS analysis

ROS in cultured cells were measured by incubating cells with 5 μM CM-H2DCFDA (C6827, Life Technologies) for 30 min at 37°C. DCF fluorescence was acquired on the Attune NxT (ThermoFisher) flow cytometer and analyzed using FlowJo software (Tree Star, Ashland, Oregon).

## Mitochondrial respiration

Oxygen consumption rate (OCR) experiments were performed using the XFe-96 apparatus from Seahorse Bioscience (Agilent, Santa Clara, California). Cells were seeded to ~80% confluence in at least triplicate for each condition in RPMI. The following day, wells were treated with 250 nM CB-839 for 4 hr. Media was completely replaced with reconstituted RPMI with 12.5 mM glucose and 2 mM glutamine (no sodium bicarbonate) adjusted to pH 7.4 and incubated for 45 min at 37°C in a $CO_2$-free incubator prior to measurements. Maximal respiration measurements were obtained by performing a mito-stress test. Briefly, cells were prepared as described above. After reaching steady-state respiratory flux (basal respiration), 100 μM oligomycin was injected to inhibit ATP synthase. Next, respiration was uncoupled from oxidative phosphorylation by addition of carbonylcyanide p-trifluoromethoxyphenylhydrazone (FCCP) (Sigma)until maximal respiratory capacity was reached

(up to 2 μM in 0.5 μM increments). Finally, respiration was inhibited by the addition of 50 μM rotenone and 50 μM antimycin A (Sigma).

## Animal experiments

All animal studies described were approved by the NYU Langone Medical Center Institutional Animal Care and Use Committee. $1 \times 10^6$ cells were implanted subcutaneously into NSG mice. After tumor establishment phase (13 days post implantation), animals were randomized and assigned to the following groups: vehicle treated and doxycycline diet, vehicle treated and normal diet, CB-839 treated and doxycycline diet, or CB-839 treated and normal diet. Animals were treated with 200 mg/kg CB-839 or vehicle (Calithera, San Francisco, California) twice daily administered through oral gavage as before (*Davidson et al., 2016*). The vehicle contained 25% (w/v) hydroxypropyl-β-cyclodextrin in 10 mmol/L citrate (pH 2.0), and CB839 was formulated at 20 mg/mL for a final dosing volume of 10 mL/kg.

## Glutamate secretion

Extracellular glutamate was measured by collecting fresh and spent medium after 24 hr of growth. Cells were assumed to grow exponentially over the culture period. Media was analyzed using a YSI biochemistry analyzer (Yellow Springs Instruments, Yellow Springs, Ohio).

## shRNA and sgRNA cloning and cell line generation

Doxycyline-induced knockdown of xCT was achieved by cloning miR-E shRNAs targeting *Slc7a11* into the LT3GEPIR vector as previously described in detail (*Fellmann et al., 2013*). Briefly, LT3GEPIR was digested with XhoI and EcoRI, and purified with a gel extraction kit (Qiagen). Single stranded ultramers were amplified with forward primer miRE-XhoI (5'- TGAACTCGAGAAGGTATATTGCTG TTGACAGTGAGCG-3') and reverse primer miRE-EcoRI (5'-TCTCGAATTCTAGCCCCTTGAAG TCCGAGGCAGTAGGC-3'). Amplicons were gel purified, digested with XhoI and EcoRI, cleaned up with PCR purification kit (Qiagen) and ligated into the cut LT3GEPIR vector with T4 DNA Ligase at a 3:1 insert:vector molar ratio. Vectors were transduced into cells and selected with 3 and 6 ug/ml puromycin for three plus three days. Knockdown of xCT/*Slc7a11* was verified by western blot and qPCR analysis following 72 hr of treatment with 1 ug/ml doxycycline. Cell lines transduced with the first two efficient shRNAs (#1 and #3), along with a non-efficient shRNA as control (#5), were used in the study. Sequences of ultramers obtained from Integrated DNA Technologies (Coralville, Iowa):

#1 - TGCTGTTGACAGTGAGCGCCAGGAAGAGACACAAGTCTAATAGTGAAGCCACAGATGTATTA-GACTTGTGTCTCTTCCTGATGCCTACTGCCTCGGA

#2 - TGCTGTTGACAGTGAGCGCAAGGAAGATTTTAGTTATTAATAGTGAAGCCACAGATGTATTAA TAACTAAAATCTTCCTTTTGCCTACTGCCTCGGA

#3 – TGCTGTTGACAGTGAGCGCCAGGAGCTTTATGTAAATGTATAGTGAAGCCACAGATGTATACA TTTACATAAAGCTCCTGATGCCTACTGCCTCGGA

#5 – TGCTGTTGACAGTGAGCGCCAGATTATACTAGAAGTTGTATAGTGAAGCCACAGATGTA TACAACTTCTAGTATAATCTGTTGCCTACTGCCTCGGA

#6 – TGCTGTTGACAGTGAGCGCTAAGTTCTATGTGATACAGAATAGTGAAGCCACAGATGTATTCTG TATCACATAGAACTTATTGCCTACTGCCTCGGA

#7 – TGCTGTTGACAGTGAGCGCAGGAACTATGTTTGTACTAAATAGTGAAGCCACAGATGTATTTAG TACAAACATAGTTCCTATGCCTACTGCCTCGGA

#8 - TGCTGTTGACAGTGAGCGCCCAGAAGACTCTAAAGAATTATAGTGAAGCCACAGATGTATAA TTCTTTAGAGTCTTCTGGTTGCCTACTGCCTCGGA

Transcriptional activation of various genes was achieved by cloning sgRNAs targeting the region upstream of the transcriptional start sites for *Slc7a11*, *Gclc*, and *Gclm* into the lenti sgRNA(MS2)

_puro and zeo backbones (Addgene plasmid #73797 and 61247, respectively) as previously described (*Cong et al., 2013*). Briefly, both backbones were digested with Bsmb1 (New England Biosciences, Ipswich, Massachusetts) and purified with a gel extraction kit (Qiagen). Single stranded oligo's were annealed and phosphorylated using T4 PNK (New England Biosciences). Phosphorylated and annealed oligos were then ligated into the purified digested backbones using Quick Ligase (New England Biosciences). Vectors were transduced into KP cells already containing the other two component vectors for the SAM system and selected with 3 and 6 ug/ml puromycin for three plus three days. Transcriptional activation was subsequently verified by qPCR. Sequences of oligos obtained from Integrated DNA Technologies:

sg*Slc7a11*:
sense- CACCGCCTGTCACACCAACTTACT
antisense- AAACAGTAAGTTGGTGTGACAGGC
sg*Gclc:*
sense- CACCGAATAAGGACTGAAAAGTCT
antisense- AAACAGACTTTTCAGTCCTTATTC
sg*Gclm*:
sense: CACCGGTGACTGGCCCTCGGCCTG
antisense: AAACCAGGCCGAGGGCCAGTCACC

## GC/MS analysis of polar metabolites and stable isotope tracing

For glutamine tracing, $2 \times 10^5$ cells were seeded in 2 mL of RPMI-1640 in 6 well plates. Where indicated, cells were pretreated for 12 hr with 1 μM of KI696. Media was then replaced with 2 mL of fresh RPMI-1640 containing 2 mM of [U-$^{13}$C]-L-glutamine (Cambridge Isotope Laboratory, Tewksbury, Massachusetts) and 1 μM of KI696 where indicated. For PC flux analysis, $1 \times 10^5$ cells were seeded in 1 mL of RPMI Cells were cultured for 8 hr to reach steady state labeling of TCA cycle intermediates. Cells were washed 1X in ice cold saline and then collected by scraping in 600 uL of 80% (v/v) of ice cold methanol containing 1.4 ug/mL norvaline (Sigma Aldrich). Samples were vortexed for 10 min at 4°C and then centrifuged at max speed for 10 min. Supernatant was transferred to fresh tubes and then dried under nitrogen. Dried and frozen metabolite extracts were then derivatized with 16 uL of MOX reagent (ThermoFisher) for 60 min at 37°C and *N*-tert-butyldimethylchlorosilane (Sigma Aldrich) for 30 min at 60°C. After derivatization, samples were analyzed by GC-MS using a DB-35MS column (Agilent Technologies) in an Agilent 7890A gas chromatograph coupled to an Agilent 5997B mass spectrometer. Helium was used as the carrier gas at a flow rate of 1.2 mL/minute. One microliter of sample was injected in split mode (split 1:1) at 270°C. After injection, the GC oven was held at 100°C for 1 min and then increased to 300°C at 3.5 °C/minute. The oven was then ramped to 320°C at 20 °C/minute and held for 5 min at 320°C.

The MS system operated under electron impact ionization at 70 eV and the MS source and quadrupole were held at 230°C and 150°C respectively. The detector was used in scanning mode, and the scanned ion range was 10–650 m/z. Mass isotopomer distributions were determined by integrating appropriate ion fragments for each metabolite (*Lewis et al., 2014*) using MatLab (Mathworks, Natick, Massachusetts) and an algorithm adapted from Ferandez and colleagues (*Fernandez et al., 1996*) that corrects for natural abundance.

### Statistics

Values are presented as mean ± SEM. For statistical analyses, we used Graphpad Prism software v.7.02 (La Jolla, California): one-way ANOVA with Tukey's post-hoc test for cell growth assays, OCR measurements, Glutathione levels and TCA Metabolites in *Figure 4e*; Mann-Whitney test for intra and extra cellular Glutamate levels and [U-$^{13}$C]-L glutamine labeled metabolites in *Figure 3e*; 2-sided Student's t-test for gene expression analyses and TCA Metabolites in *Figure 3c*; and two-way ANOVA for tumor growth. All experiments were repeated at least twice unless otherwise stated. All *n* indicate biological replicates.

## Acknowledgements

We would like to thank Calithera Biosciences for providing CB-839 and Vehicle used in the in vivo experiments. We would also like to thank Elizavet Freinkman for assistance with LC-MS/MS analysis

via the Whitehead Institute Metabolite Profiling Core Facility. TP is supported by the NIH (K22CA201088-01). VIS received support from the Swedish Medical Research Council, the Wenner-Gren Foundation and is the recipient of EMBO Long Term Fellowship ALTF 1451–2015 co-funded by the European Commission (LTCOFUND2013, GA-2013–609409) with support from Marie Curie Actions. SEL is supported by an NIH training grant (5T32HL007151-38).

## Additional information

### Funding

| Funder | Grant reference number | Author |
| --- | --- | --- |
| National Institutes of Health | K22CA201088-01 | Thales Papagiannakopoulos |
| European Molecular Biology Organization | GA-2013-609409 | Volkan I Sayin |
| National Institutes of Health | 5T32HL007151-38 | Sarah E LeBoeuf |

The funders had no role in study design, data collection and interpretation, or the decision to submit the work for publication.

### Author contributions

Volkan I Sayin, Conceptualization, Data curation, Formal analysis, Supervision, Funding acquisition, Investigation, Visualization, Writing—original draft, Project administration, Writing—review and editing, Designed the study and wrote the manuscript with comments from all the authors; Sarah E LeBoeuf, Conceptualization, Resources, Data curation, Formal analysis, Supervision, Validation, Investigation, Visualization, Methodology, Writing—original draft, Project administration, Writing—review and editing, Designed the study, performed and analyzed experiments experiments, and wrote the manuscript with input from all the authors; Simranjit X Singh, Conceptualization, Data curation, Formal analysis, Supervision, Validation, Investigation, Visualization, Methodology, Writing—original draft, Project administration, Writing—review and editing, Designed the study, performed and analyzed experiments experiments, and wrote the manuscript with input from all the authors; Shawn M Davidson, Warren L Wu, Data curation, Formal analysis, Writing—review and editing, Performed and analyzed experiments; Douglas Biancur, Data curation, Formal analysis, Methodology, Writing—review and editing, Performed and analyzed experiments and provided feedback and interpretation of metabolism data; Betul S Guzelhan, Data curation, Formal analysis, Validation, Visualization, Writing—review and editing, Performed and analyzed experiments; Samantha W Alvarez, Julian Ubriaco, Data curation, Formal analysis, Performed and analyzed experiments; Triantafyllia R Karakousi, Data curation, Formal analysis, Validation, Writing—review and editing, Performed and analyzed experiments; Anastasia Maria Zavitsanou, Data curation, Formal analysis, Visualization, Writing—review and editing, Performed and analyzed experiments; Alexander Muir, Patrick J Morris, Data curation, Performed and analyzed experiments; Dimitris Karagiannis, Data curation, Software, Methodology, Performed and analyzed experiments and provided feedback and interpretation of metabolism data; Craig J Thomas, Resources, Methodology, Provided KI696; Richard Possemato, Resources, Methodology, Writing—review and editing, Provided KI696; Matthew G Vander Heiden, Resources, Supervision, Methodology, Writing—review and editing; Thales Papagiannakopoulos, Resources, Supervision, Investigation, Methodology, Project administration, Writing—review and editing, Provided feedback and interpretation of metabolism data

### Author ORCIDs

Sarah E LeBoeuf http://orcid.org/0000-0003-1580-6536
Matthew G Vander Heiden https://orcid.org/0000-0002-6702-4192
Thales Papagiannakopoulos http://orcid.org/0000-0002-2251-1624

### Ethics

Animal experimentation: All animal studies described were approved by the NYU Langone Medical Center Institutional Animal Care and Use Committee in accordance with protocol #170605.

### Decision letter and Author response

Decision letter https://doi.org/10.7554/eLife.28083.017
Author response https://doi.org/10.7554/eLife.28083.018

---

## Additional files

### Supplementary files

• Transparent reporting form
DOI: https://doi.org/10.7554/eLife.28083.016

---

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
