## [Decision Letter]

Thank you for submitting your article "Activation of the NRF2 antioxidant program causes defects in central carbon metabolism" for consideration by *eLife*. Your article has been reviewed by three peer reviewers, one of whom is a member of our Board of Reviewing Editors, and the evaluation has been overseen by Sean Morrison as the Senior Editor. The following individual involved in review of your submission has agreed to reveal his identity: Oliver DK Maddocks (Reviewer #3).

The reviewers have discussed the reviews with one another and the Reviewing Editor has drafted this decision to help you prepare a revised submission.

Summary:

These authors studied metabolic reprogramming downstream of activating the Nrf2 transcriptional program in cancer cells. Nrf2 is negatively regulated by Keap1, and gain of function mutations in this pathway occur in lung cancer. The authors find that mutating Keap1 sensitizes cultured cells to glutamine deprivation and to glutaminase inhibitors. The mechanism involves enhanced glutamate export via SLC7A11, which allows cells to import cystine to meet demands for glutathione biosynthesis. Inhibiting SLC7A11 reverses sensitivity to glutaminase inhibition in culture and in tumors. The authors go on to provide evidence that Nrf2 activation imposes an addiction to anaplerosis from glutamine, such that TCA cycle metabolism declines when glutaminase is inhibited; these cells can be rescued by providing alternative anaplerotic precursors. Importantly, pharmacological activation of the Nrf2 pathway is sufficient to addict cells to glutamine. Thus, the authors propose a balance between glutamine anaplerosis and cysteine-based antioxidant production that could provide a therapeutic window for glutaminase inhibitors.

Overall, the reviewers found the paper to be solid, clearly presented, and focused on an interesting topic. With glutaminase inhibitors in clinical trials but lacking enrollment biomarkers, the main findings of the paper could have translational relevance. However, the reviewers also identified a few areas where the paper could be improved.

Essential revisions:

1) In the context of this paper there are three main ways that glutamate is utilized: (1) TCA cycle, (2) cystine/glutamate exchange, and (3) GSH synthesis. The sensitivity of KPK cells to glutaminase inhibition is rescued by providing TCA cycle precursors (pyruvate, DMG), or by blocking cystine/glutamate exchange, whereas anti-oxidants (trolox & NAC) fail to rescue CB839 sensitivity. This is taken as evidence that GSH synthesis is the least important of the three glutamate sinks. However, a better way to test this is to diminish GSH synthesis directly (e.g. use intermediate doses of BSO in KPK cells to decrease GSH synthesis to KP cell levels).

2) It would be useful to test if adding TCA cycle precursors (pyruvate/DMG) to KPK cells under basal conditions (i.e. without any inhibitors present) increases TCA cycle metabolite levels and oxygen consumption. This would confirm that glutamate availability is genuinely a metabolic "bottleneck" for TCA cycle/OXPHOS in these cells.

3) Nrf2 activation itself does not impede proliferation under normal conditions despite diminishing anaplerosis/TCA cycle/OXPHOS. If this is the case then an overall interpretation of the paper could be that the KPK cells sacrifice excess TCA cycle capacity for excess anti-oxidant capacity, which might be an appealing trade-off during tumorigenesis. This is worth clarifying because it would help explain selection for Nrf2-activating mutations, and would help differentiate this paper from previous work (e.g. PMID: 28429737, 22789539). Do KPK cells have lower ROS levels than KP cells under basal conditions? Are KPC cells more resistant to elevated ROS levels (e.g. H_2_O_2_) than KP?

4) Why does pyruvate rescue cells subjected to CB-839 or glutamine deprivation? Pyruvate is ostensibly not limiting under these conditions, and adding more of it does not rescue OCR. The implication is that pyruvate provides an alternative anaplerotic substrate via pyruvate carboxylase, but this should be validated.

5) p53 is known to inhibit cystine transporter activity (Jiang L et al., Nature 520, 57-62). Does adding cisplatin to activate p53 in both KP and KPK cells alter sensitivity to glutaminase inhibition? If so, this would broaden the impact of the findings.

---

## [Author Response]

Essential revisions:1) In the context of this paper there are three main ways that glutamate is utilized: (1) TCA cycle, (2) cystine/glutamate exchange, and (3) GSH synthesis. The sensitivity of KPK cells to glutaminase inhibition is rescued by providing TCA cycle precursors (pyruvate, DMG), or by blocking cystine/glutamate exchange, whereas anti-oxidants (trolox & NAC) fail to rescue CB839 sensitivity. This is taken as evidence that GSH synthesis is the least important of the three glutamate sinks. However, a better way to test this is to diminish GSH synthesis directly (e.g. use intermediate doses of BSO in KPK cells to decrease GSH synthesis to KP cell levels).

To address this point we cultured KPK cells in decreasing doses of BSO, which decreased GSH synthesis and total GSH levels in a dose dependent manner (Figure 3—figure supplement 1). When KPK cells are treated with CB-839 in the presence of BSO, at concentrations that reduce the total GSH levels to KP cell levels or lower, we observe no change in sensitivity to glutaminase inhibition (Figure 3—figure supplement 1). These experiments directly address essential comment 1, suggesting that GSH synthesis is the least important glutamate sink.

To further address essential Comment 1, we took a genetic approach. We used the CRISPR synergistic activation mediator system (SAM CRISPRa) (Figure 4), which takes advantage of the DNA targeting capabilities of a catalytically dead Cas9 protein fused to VP64 and a modified sgRNA that recruits and targets transcriptional activators to promoter regions upstream of transcriptional start sites of genes of interest to enhance endogenous transcription. Utilizing SAM CRISPRa, we increased expression of *Gclc* and *Gclm*, enzymes involved in the rate-limiting step of GSH synthesis, as well as *Slc7a11* either alone or in combination with *Gclc* or *Gclm* in KP cells. In all cases we were able to increase expression of the targeted gene (Figure 4). Additionally, increased gene expression of individual targets or in combination increased total GSH levels (Figure 4—figure supplement 1). However, increased GSH production per se in these cell lines did not result in increased sensitivity to CB-839 treatment. We did observe an increase in CB-839 sensitivity in KP SAM CRISPRa cells with double guides (sg*Slc7a11* and sg*Gclm*/sg*Gclc*) (Figure 4), however, the sensitivity was not equivalent to what we observed in KPK cells.

These results further emphasize that GSH synthesis is the least important glutamate sink and also highlight that altering GSH levels alone is not sufficient to recapitulate CB-839 sensitivity.

While SAM CRISPRa induces transcription of genes from endogenous loci, levels of gene induction and GSH levels could not be induced to levels equivalent to KPK cells (3-10 fold vs 10-100 fold gene expression and 1.5-2 fold vs 3 fold GSH levels in KP SAM CRISPRa and KPK cells respectively). This difference in induction of gene expression could explain the limited increase in sensitivity to CB-839 seen in KP SAM CRISPRa cells compared to KPK cells. However, Nrf2 is a pleiotropic transcription factor that is known to regulate a plethora of antioxidant genes. Therefore, we cannot exclude the possibility that induction of additional genes (including other antioxidants such as Thioredoxin) resulting from constitutive active Nrf2 signaling are required to fully recapitulate the metabolic rewiring resulting in robust sensitivity to CB-839 in KPK cells.

We have added a paragraph in the Discussion section to discuss and address these points.

2) It would be useful to test if adding TCA cycle precursors (pyruvate/DMG) to KPK cells under basal conditions (i.e. without any inhibitors present) increases TCA cycle metabolite levels and oxygen consumption. This would confirm that glutamate availability is genuinely a metabolic "bottleneck" for TCA cycle/OXPHOS in these cells.

Please see supplemental Figure 3 and Figure 3—figure supplement 3 and B to address this point. We have measured the levels of TCA cycle intermediates in KP and KPK cells after treatment with 2mM pyruvate or 2mM DMG. We see that addition of either metabolite increases the relative abundance of TCA cycle intermediates in KPK cells. The increase in TCA cycle metabolites we observe in KPK cells is either equivalent to or in excess of the levels of these intermediates in KP cells in normal media conditions (Figure 3). Additionally, both pyruvate and DMG supplementation elevate basal levels of OCR in the presence of CB-839 (Figure 3). However, in normal media conditions, without the addition of CB-839, the supplementation of either pyruvate or DMG does not result in a significant increase in OCR in KPK cells (Figure 3—figure supplement 3). We believe that is a result of reduced reserve respiratory capacity in KPK cells as compared to KP cells (Figure 3—figure supplement 3). Basal OCR is ~75% of maximal OCR in KPK cells. It is likely more difficult to increase basal respiration in these cells since mitochondria are functioning at close to maximal capacity whereas in KP cells, there is a greater reserve respirator capacity (Figure 3—figue supplement B) and we do see that supplementation with either pyruvate or DMG significantly increases OCR (Figure 3—figure supplement 3).

3) Nrf2 activation itself does not impede proliferation under normal conditions despite diminishing anaplerosis/TCA cycle/OXPHOS. If this is the case then an overall interpretation of the paper could be that the KPK cells sacrifice excess TCA cycle capacity for excess anti-oxidant capacity, which might be an appealing trade-off during tumorigenesis. This is worth clarifying because it would help explain selection for Nrf2-activating mutations, and would help differentiate this paper from previous work (e.g. PMID: 28429737, 22789539). Do KPK cells have lower ROS levels than KP cells under basal conditions? Are KPC cells more resistant to elevated ROS levels (e.g. H_2_O_2_) than KP?

We have clarified this point clearly in the Discussion. In addition, we assessed ROS levels and sensitivity to ROS inducing agents like the reviewers suggested. Basally, KPK cells have much lower ROS levels compared to KP cells (Figure 1—figure supplement 1) and are indeed more resistant to oxidative stress, exemplified with treatment with H_2_O_2_ (Figure 1) as well as other oxidizing agents such as galactose oxidase (Figure 1) and menadione (Figure 1—figure supplement 1).

4) Why does pyruvate rescue cells subjected to CB-839 or glutamine deprivation? Pyruvate is ostensibly not limiting under these conditions, and adding more of it does not rescue OCR. The implication is that pyruvate provides an alternative anaplerotic substrate via pyruvate carboxylase, but this should be validated.

Pyruvate supplementation does modestly, but significantly increase OCR in the presence of CB-839 (Figure 3). However, it is likely that there is increased carbon flux through pyruvate carboxylase in efforts to continue to fuel the TCA cycle when glutamate availability is limited by treatment with CB-839. To address this issue, we performed tracing experiments using stable-isotope labeled glucose (3C^13^-D-Glucose) to look at PC flux in normal media conditions as well as labeled pyruvate (1C^13^-Pyruvate) to look at PC flux in the context of pyruvate rescue. We observe that treatment with CB-839 increases PC flux in normal media as well as when media is supplemented with pyruvate (Figure 3). This suggests that when glutamate availability is limited cells divert more glucose-derived carbon through PC to help replenish the TCA cycle.

5) p53 is known to inhibit cystine transporter activity (Jiang L et al., Nature 520, 57-62). Does adding cisplatin to activate p53 in both KP and KPK cells alter sensitivity to glutaminase inhibition? If so, this would broaden the impact of the findings.

Inactivation of p53 occurs in a large percentage of human tumors. The mouse model from which our KP and KPK cell lines are derived from models lung cancer through the constitutive activation of Kras by expression of the endogenous Kras^G12D^ allele and concomitant loss of p53. Additionally, the human LUAD cell lines as well as cell lines from other cancer subtypes used throughout this paper are also p53 mutants preventing us from utilizing strategies to activate p53 in the cell lines in this manuscript. Despite this, we tested the differential sensitivity to cisplatin between KP and KPK cells and saw no difference. In addition there was no difference in CB-839 sensitivity in combination with cisplatin in both KP and KPK cells (data not shown). As the initial question raised by reviewers was not possible to be addressed in our system, we felt the data not relevant to include in the revised version of the manuscript.